# A localized view on molecular dissociation via electron-ion partial covariance

Felix Allum [1,2✉], Valerija Music [3,4,5], Ludger Inhester [6✉], Rebecca Boll [4], Benjamin Erk [5], Philipp Schmidt [3,4], Thomas M. Baumann[4], Günter Brenner[5], Michael Burt [1], Philipp V. Demekhin [3], Simon Dörner[5], Arno Ehresmann [3], Andreas Galler[4], Patrik Grychtol[4], David Heathcote [1], Denis Kargin [7], Mats Larsson[8], Jason W. L. Lee [1,5], Zheng Li [6,9], Bastian Manschwetus[5], Lutz Marder [3], Robert Mason[1], Michael Meyer [4], Huda Otto[3], Christopher Passow[5], Rudolf Pietschnig [7], Daniel Ramm[5], Kaja Schubert[5], Lucas Schwob[5], Richard D. Thomas [8], Claire Vallance [1], Igor Vidanović[7], Clemens von Korff Schmising[10], René Wagner[4], Peter Walter [11], Vitali Zhaunerchyk[12], Daniel Rolles [13], Sadia Bari [5], Mark Brouard [1] & Markus Ilchen[3,4,5✉]

Inner-shell photoelectron spectroscopy provides an element-specific probe of molecular structure, as core-electron binding energies are sensitive to the chemical environment. Short-wavelength femtosecond light sources, such as Free-Electron Lasers (FELs), even enable time-resolved site-specific investigations of molecular photochemistry. Here, we study the ultraviolet photodissociation of the prototypical chiral molecule 1-iodo-2-methylbutane, probed by extreme-ultraviolet (XUV) pulses from the Free-electron LASer in Hamburg (FLASH) through the ultrafast evolution of the iodine 4d binding energy. Methodologically, we employ electron-ion partial covariance imaging as a technique to isolate otherwise elusive features in a two-dimensional photoelectron spectrum arising from different photo-fragmentation pathways. The experimental and theoretical results for the time-resolved electron spectra of the $4d_{3/2}$ and $4d_{5/2}$ atomic and molecular levels that are disentangled by this method provide a key step towards studying structural and chemical changes from a specific spectator site.

[1] The Chemistry Research Laboratory, Department of Chemistry, University of Oxford, Oxford OX1 3TA, UK. [2] Stanford PULSE Institute, SLAC National Accelerator Laboratory, 2575 Sand Hill Road, Menlo Park, CA 94025, USA. [3] Institut für Physik und CINSaT, Universität Kassel, Heinrich-Plett-Straße 40, D-34132 Kassel, Germany. [4] European XFEL, Holzkoppel 4, 22869 Schenefeld, Germany. [5] Deutsches Elektronen-Synchrotron DESY, Notkestr. 85, 22607 Hamburg, Germany. [6] Center for Free-Electron Laser Science CFEL, Deutsches Elektronen-Synchrotron DESY, Notkestr. 85, 22607 Hamburg, Germany. [7] Institut für Chemie, Universität Kassel, Heinrich-Plett-Straße 40, D-34132 Kassel, Germany. [8] Stockholm University, AlbaNova University Center, 114 21 Stockholm, Sweden. [9] State Key Laboratory for Mesoscopic Physics, School of Physics, Peking University, Beijing 100871, China. [10] Max Born Institute, Max-Born-Straße 2A, 12489 Berlin, Germany. [11] SLAC National Accelerator Laboratory, 2575 Sand Hill Road, Menlo Park, CA 94025, USA. [12] University of Gothenburg, 405 30 Gothenburg, Sweden. [13] J. R. Macdonald Laboratory, Physics Department, Kansas State University, 1228 Martin Luther King Jr. Dr., Manhattan, KS 66506, USA. ✉email: fallum@stanford.edu; ludger.inhester@desy.de; markus.ilchen@desy.de

Molecular restructuring and its consequences for molecular function are of ubiquitous interest across a variety of scientific disciplines. The involved physical and chemical dynamics typically progress on the femtosecond timescale, which can be observed in "real-time" through a range of ultrafast spectroscopic techniques[1]. Modern technological developments in high-intensity short-wavelength FELs have extended such methods for probing ultrafast chemistry in a site-selective manner by utilizing wavelengths of light which can selectively address core orbitals[2–9].

Ultrafast molecular fragmentation can cause significant core-electron-binding energy changes. These changes are typically on the order of few eV for chemical shifts of neutral fragments, tens of eV for delocalized charges in the valence shell, and more than a hundred eV for localized core holes[10]. Such shifts are measurable by photoelectron spectroscopy and can be used to study photochemistry in real-time from a specific observer site[5–7]. An often limiting factor of such studies is that it is difficult to distinguish smaller shifts from static signal originating from ground-state molecules and background[7]. In addition, relating delay-dependent signal to a specific underlying process is challenging, particularly in the case of more complex molecules which may undergo a range of photochemical processes following photoexcitation or ionization. One potential solution to overcome this limitation is to utilize electron–ion correlations, allowing electron spectroscopy to be applied in a channel-resolved manner, by isolating contributions in an electron spectrum correlated to a specific photofragmentation channel, determined by ion spectroscopy[11–13]. Electron–ion coincidence techniques have proven to be very powerful, but are limited to very low count rates, such that multiple particles produced in the same laser pulse can be assigned to a single event[14,15]. While progress in coincidence experiments at FELs has been made[2,9,16–18], sufficient data collection rates for highly differential insights into molecular fragmentation channels still pose a considerable technical challenge that can prospectively be tackled by high-repetition-rate FELs[19]. Here, we exploit an alternative method to determine charged-particle correlations at far higher count rates per photon pulse; through calculating the covariance, a measure of linear correlation between the signals of interest recorded over many data acquisition cycles (i.e., laser shots)[20–22]. This holds the promise of being applicable even to larger molecules[23]. Although the inherently unstable conditions due to stochastic pulse generation at self-amplified spontaneous emission (SASE) FELs provide challenges for correlation techniques, schemes have been developed to not only correct for the adverse effects of such fluctuations but effectively exploit them through either partial[24,25] or contingent[26] covariance analysis. In this work, we demonstrate the extension of these techniques, usually applied to a 1D mass spectrum, to a 3D Velocity-Map Imaging (VMI) study of the ultrafast evolution of electronic structure during a photodissociation, at a particular core site, in a channel-resolved manner.

We use this technique in order to investigate the properties and dynamics of the prototypical chiral molecule 1-iodo-2-methylbutane ($C_2H_5CH(CH_3)CH_2I$) at the iodine 4d edge, as it is a prominent candidate for approaching dynamical investigations of chirality with FELs in future studies. Understanding and benchmarking the underlying ultrafast photochemistry is an important prerequisite for these kind of studies. In particular, we excite the molecule in the ultraviolet (UV) and predominantly trigger neutral dissociation at its carbon–iodine bond (shown schematically in Fig. 1). We demonstrate the value of partial covariance analysis for following the iodine in its dynamical change from molecular to isolated atomic environments through channel-resolved photoelectron spectroscopy.

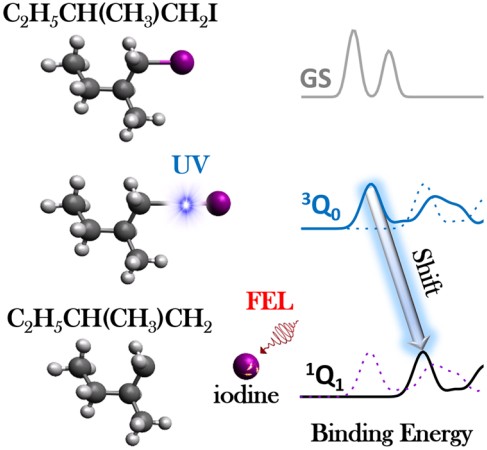

**Fig. 1 Schematic representation of the experimental scheme to study the ultrafast photodynamics of 1-iodo-2-methylbutane.** Photoexcitation (predominantly to the excited state of $^3Q_0$ symmetry) is initiated by a UV pump pulse. The photoexcited molecule is interrogated at a series of pump-probe delays by a XUV-FEL pulse, probing the photoelectron binding energy of the I (4d) core orbital. Measured changes in the binding energy during the photodissociation can be related to the underlying photochemistry, supported by quantum simulations of the photoionization process.

## Results and discussion

Samples of R/S-1-iodo-2-methylbutane were introduced as a continuous molecular beam, seeded in helium, into the CAMP endstation[27] at the beamline BL1 of FLASH1[28] at DESY in Hamburg, Germany. The molecule was dissociated at its C–I bond following single-photon UV excitation (267 nm (4.6 eV), ~150 fs, maximum pulse energy of 7 µJ). As is the case in alkyl iodides in general[29], photoabsorption in this region arises due to excitation from the iodine lone pair ($n_I$) to the C–I antibonding ($\sigma^*$) orbital. The evolving chemical dynamics following photoexcitation are investigated from the viewpoint of the released neutral iodine atom via a time-delayed, ~63.5 eV, FEL-based probe pulse with ~50 fs duration and pulse energy at the target of about 1 µJ (see "Methods" for details). Due to the large cross-section difference to other constituents and electronic orbitals, the I 4d orbital is predominantly ionized[30]. The photoions and photoelectrons produced are velocity-mapped to a pair of position-sensitive detectors[27,31–33] (as described in more detail in "Methods"). By using partial covariance analysis to select only electrons that are emitted from neutrally dissociated iodine, and following their time evolution during the photolysis, an advanced scheme for femtochemistry is enabled. The interpretation of the delay-dependent photoelectron spectra is supported by state-of-the-art simulations of photoionization[34].

**Velocity-map ion imaging.** Figure 2a shows mass spectra of 1-iodo-2-methylbutane exposed to the UV and XUV pulses alone, or with both pulses for positive pump-probe delays (UV preceding the XUV). At the employed intensities, very little multi-photon dissociative ionization is initiated by the UV pulse alone, whereas the XUV pulse causes extensive ionic break-up. In the two-color experiment, a clear pump-probe signal can be observed most prominently in the $I^{2+}$ ion, whose yield is significantly enhanced when the UV pulse precedes the XUV. As ionization at the I 4d orbital by the XUV predominantly results in two charges after Auger decay[30], UV-induced neutral photodissociation followed by ionization at the nascent iodine atoms by the XUV would lead to an enhanced $I^{2+}$ signal at sufficiently large internuclear distances, for which charge transfer does not occur[3].

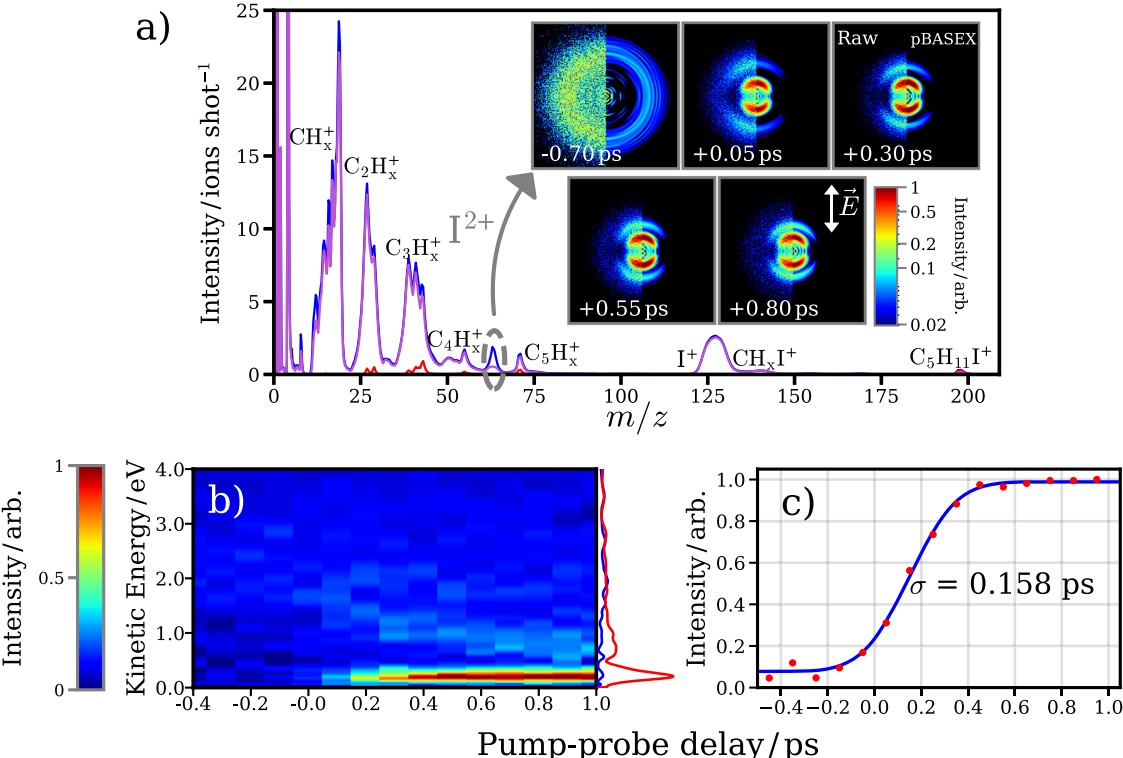

**Fig. 2 Summary of the time-resolved ion-imaging results. a** Normalized ion mass spectra recorded for 1-iodo-2-methylbutane with the XUV only (magenta), UV only (red) and UV-early-XUV late at a pump-probe delay of +0.80 ps (blue). Inset: Raw (left) and reconstructed (right)[78] velocity-map $I^{2+}$ ion images at a series of pump-probe delays. The −0.70 ps image's intensity has been multiplied by 5, to increase the visibility of the weak, high-KER channel. The polarization axis of the UV laser is vertical in these images. **b** Delay-dependent kinetic-energy distribution for the $I^{2+}$ ion. The UV-early (>1 ps delay) and UV late (<−0.2 ps delay) distributions are projected in red and blue respectively. **c** Integrated yield of the low kinetic-energy (<0.4 eV) feature as a function of pump-probe delay (red points) with a fit to a normal cumulative distribution function (blue line).

Small enhancements of other fragments are also visible in comparison to the XUV-only spectrum.

Velocity-map images for the $I^{2+}$ ion at a series of pump-probe delays, shown in the inset of Fig. 2a, provide insight into the UV-induced C–I dissociation. At negative pump-probe delays (UV late), a weak, broad feature at high radii is observed, that is assigned to a (multiphoton) XUV-induced Coulomb explosion of the parent molecule. When the UV pulse pre-excites the molecules, two clear features emerge in the ion images. First, there is a strong contribution at low radii, which is peaked along the UV polarization axis. As is expected for one-photon transitions, the intensity of this feature as a function of angle to the polarization axis, $I(\theta)$, is of the form $I(\theta) = (\sigma/4\pi)[1 + \beta P_2(\cos\theta)]$, where $P_2$ is the second Legendre polynomial, and $\beta$ is the anisotropy parameter. This takes limiting values of −1 and +2 for transitions of pure perpendicular or parallel nature, respectively (under the assumption of a prompt photodissociation)[35–37]. In this case, a $\beta$ value of ≈1.80 is extracted. This is expected for neutral photodissociation following a parallel excitation predominantly to the $^3Q_0$ state, as observed in similar alkyl iodides[29]. The delay-dependent $I^{2+}$ kinetic energy plotted in Fig. 2b and c shows the integrated intensity of the low kinetic energy, neutral dissociation feature. This signal rises on an ultrafast (few hundred fs) timescale, as is expected for a direct dissociation, as observed previously in related alkyl iodides[29]. The rise in intensity of this feature is somewhat delayed with respect to time zero. This is to be expected as, at sufficiently early pump-probe delays, charge transfer can occur between the multiply charged iodide ion produced following XUV ionization and the recoiling $C_4H_9$ radical, reducing the low energy $I^{2+}$ ions formed. Previous pump-probe studies using site-selective ionization in similar photodissociation molecules have examined differences in the delay-dependent behavior of multiple iodine charge states to extract information about distance-dependent charge-transfer probabilities[3,4,8,38]. However, in the present work, which employs a relatively weak XUV pulse which is only a few eV above the I 4d binding energy of the neutral molecule, a range of iodine ion charge states are not populated, and thus the extractable insights into charge transfer are limited and not discussed further in this manuscript.

Second, a weaker, more diffuse feature at higher radii is also visible after time zero. This moves toward the center of the image at longer pump-probe delays, indicative of a Coulombic contribution to the fragmented energy, which decreases at larger internuclear separations, i.e., longer pump-probe delays[3,4,38]. Covariance imaging analysis[38–41] confirms that this minor channel arises from a multiphoton dissociative ionization by the pump pulse, prior to XUV absorption at the iodine site. At longer pump-probe delays, the double ionization at the iodine fragment occurs when the charged alkyl co-fragment is at a greater separation, and so the Coulombic contribution to the kinetic energy of this feature decreases as pump-probe delay advances. This channel is not discussed further in the current work, which focuses on the dominant, neutral photodissociation channel. As will be demonstrated shortly, the electron–ion partial covariance imaging method used allows isolation of the photoelectron signal correlated solely to the neutral dissociation feature of interest. As shown in this section, the temporally and kinetic-energy-resolved ion-yield evolution already provides valuable information about the individual dissociation channels and allows to partially disentangle them. Deeper insights about selective contributions and processes can then be accessed by studying the underlying electronic dynamics.

**Electron–ion partial covariance imaging**. The photodissociation dynamics can be further probed through time-resolved inner-shell photoelectron spectroscopy[5–7,34] at the iodine 4d site, as demonstrated on the ultraviolet photodissociation of methyl iodide by Brauße et al.[7], in which a small increase in I 4d binding energy was detected following UV excitation. This was assigned to ionization of dissociated iodine atoms, supported by earlier synchrotron measurements of the I 4d binding energies of $CH_3I$ and $I$[42–45]. The ability to study the temporal evolution of the signal, however, was hampered by the fact that this small contribution overlaps energetically with signal arising from unpumped parent molecules (due in part to the significant FEL bandwidth); a limitation that can be tackled by the partial covariance analysis. A primary aspect of the current work is that this method can be utilized to isolate delay-dependent spectral features of interest.

Photoelectron images following irradiation of 1-iodo-2-methylbutane (seeded in He) by the UV and XUV lasers are plotted in Fig. 3a. The strong rings observed in the helium-only case are due to single- and double ionization of He by the XUV pulse, which form a significant background when 1-iodo-2-methylbutane is present, labeled "$C_5H_{11}I$+He" in Fig. 3a. Subtraction of the helium-only background image, normalized by the number of laser shots and average FEL pulse energy yields the image plotted on the right of Fig. 3a. A feature at slightly lower kinetic energy (higher binding energy) than the $He^{2+}$ photoline is observed, arising from ionization at the I 4d site in $C_5H_{11}I$. The associated electron binding energy spectrum (Fig. 3c) shows two clear peaks at approximately 56.5 eV and 58.0 eV, which can be assigned to the molecular $4d_{5/2}$ and $4d_{3/2}$ levels, respectively. In this simple association of electronic origins, the energy difference is already a reliable parameter. A higher differential view on the angular distribution patterns is not only a valuable characterization of the given electron orbital compositions of the molecule in its ground state but also potentially for the evolving composition of the chemical environment of the respective emitter site. For the static case of the electron emitted from the I 4d site of $C_5H_{11}I$, the $\beta_2$ parameter[35,46] for electrons originating from the molecular I 4d site was determined to be $\beta_2 = 0.25$ for the $4d_{5/2}$ and $\beta_2 = 0.3$ for the $4d_{3/2}$, which is in reasonable agreement to previous work on $CH_3I$[47] under the given experimental conditions and provides a benchmark for further studies. Following the time dependence of these angular distributions with partial covariance mapping during molecular dissociation is a goal for future (higher statistics) studies.

The electron velocity distributions correlated with the production of a particular photoion can be extracted by calculating the covariance between the integrated count of the ion of interest and each pixel of the electron image. As three-dimensional ion-velocity information is recorded on an shot-by-shot basis, electron spectra correlated to a specific range of ion velocities can be calculated by appropriately selecting ions within a given velocity range. Figure 3b shows the electron–ion covariance calculated for the $I^+$ ion, which is predominantly produced following the interaction of the molecule with the XUV pulse alone (see Fig. 2a). In this image, which represents the laboratory-frame photoelectron distribution correlated to the production of $I^+$ ions, the I 4d feature is clearly highlighted. However, there is still significant background present from the He seeding gas. This "false" covariance is attributed to correlations induced by the fluctuating FEL power during the experiment, which has the effect of correlating all measured signals. This can be accounted for through partial covariance analysis[24,25,48] as the FEL pulse energy is also recorded on a shot-by-shot basis[49] (details of the partial covariance procedure are given in "Methods"). An additional map, denoted the "correction" map, representing the

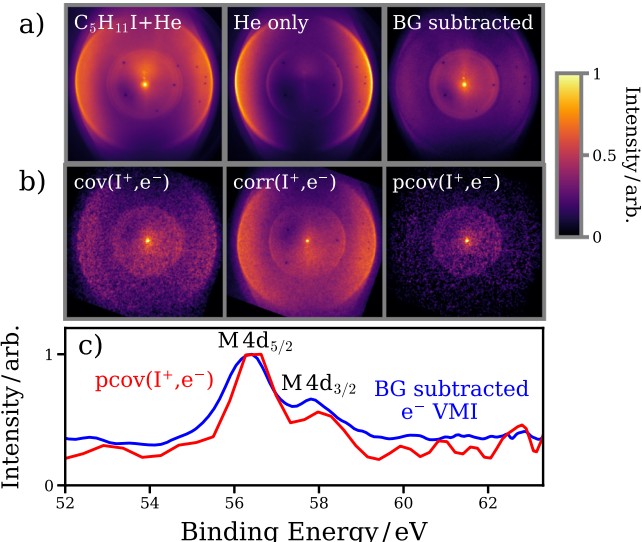

**Fig. 3 Photoelectron imaging and electron–ion partial covariance imaging. a** Photoelectron VMI images (UV-XUV) of: (left to right) 1-iodo-2-methylbutane seeded in He carrier gas, He only and 1-iodo-2-methylbutane following background subtraction. **b** Electron–ion partial covariance analysis for the $I^+$ ion, showing images of (left to right) covariance, correction term, and the partial covariance. **c** Electron spectra associated with the background-subtracted velocity-map electron image (blue) and the $I^+$ electron–ion partial covariance image (red). The two main features, arising from molecular $4d_{5/2}$ and $4d_{3/2}$ ionization, are labeled.

(linear) correlations induced by the fluctuating FEL pulse energy is constructed. Subtraction of this term from the covariance term yields the partial covariance, which isolates the true electron–ion correlations (which, in this case arise from ionizing at the I 4d orbital of 1-iodo-2-methylbutane). In panel 3c, strong principal agreement is observed between the covariant electron spectrum for $I^+$ and the equivalent spectrum obtained following subtraction of the various background contributions from the raw electron image. We note that the photoelectron spectrum extracted from the partial covariance image is noisier than the equivalent image obtained through background subtraction of the raw electron image. This is in part due to the nature of the covariance mapping procedure, which relies on statistical (Poisson) fluctuations in a noisy dataset and the detection of multiple particles, each of which have rather finite detection efficiencies. The influence of these factors on covariance mapping has been examined in detail recently[50,51]. The covariant electron spectrum importantly contains information that cannot be gleaned from the raw photoelectron spectra, namely channel-resolved information by extracting photoelectron spectra correlated to the production of a given ion channel.

As discussed previously, the low kinetic energy $I^{2+}$ ions observed in Fig. 2 are formed by a distinct pathway: UV-induced photodissociation and subsequent XUV ionization at the nascent iodine atom 4d orbital. The partial covariance image for low-velocity (i.e., originating from neutral dissociation) $I^{2+}$ ions is plotted in Fig. 4a, for long positive pump-probe delays (UV first by at least 550 fs). In Fig. 2a, a clear circular feature is observed at a significantly lower radius (higher electron binding energy) than for XUV-only ionization and fragmentation of the parent molecule. As seen in Fig. 4b, the spectrum associated with the neutral dissociation exhibits a shift to higher binding energies, by ~1.5–2 eV, consistent with synchrotron studies on the I 4d photoelectron spectra of free iodine atoms[43,44]. Crucially, and in contrast to the previous work[7], this energetic shift as a result of

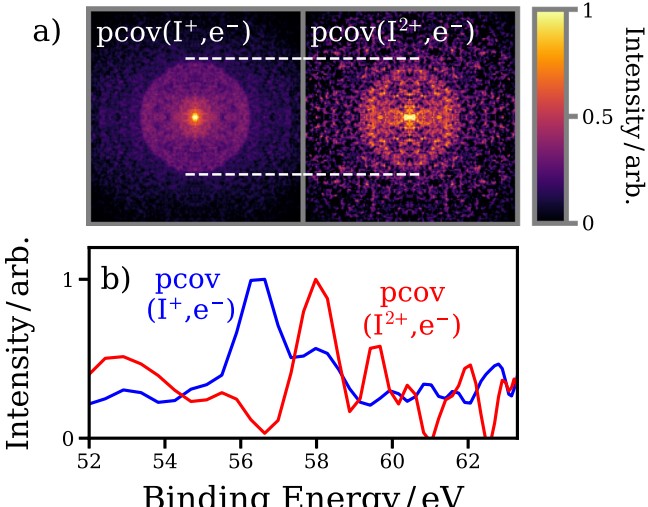

**Fig. 4 Covariant photoelectron spectra of bound and dissociated molecules. a** Electron–ion partial covariance images (symmetrized) for the $I^+$ ion and the $I^{2+}$ ion (low radius ions only, for pump-probe delays of +0.55 ps and +0.80 ps). A horizontal line at the radius of the ring seen in the $I^+$ image highlights the shift to lower radius in the $I^{2+}$ case. **b** Photoelectron spectra extracted from each of these partial covariance images.

dissociation can be completely isolated from the far stronger unpumped parent molecule signal, as well as from any competing pump-probe channels, such as the multiphoton dissociative ionization pathway. As such, the method presented here allows for decisive insights into the photochemistry of this prototypical molecule.

Figure 5 shows the covariant electron spectra associated with low-velocity $I^{2+}$ ions in a time-resolved manner. For all pump-probe delays, the electron spectra in covariance with the $I^{2+}$ photodissociation products show clear differences from the spectrum of ground-state molecules. Three main peaks can be seen in these spectra in the ~56–61 eV region, along with an immediate, unresolved, shift to higher electron binding energies. In the ~54–56 eV region, the signal (with either negative or positive intensity) is assigned to the partial covariance routine failing to correctly remove all the He-background contributions. This issue arises due to the relatively low statistics when calculating the partial covariance for a given delay bin. The peak at ~56.5 eV visible in panel 5a coincides energetically with the signal stemming from the unpumped molecule, indicated by the $I^+$ signal and displayed in more detail in Fig. 3c. At small time delays, the overall yield of the $I^{2+}$ is reduced since charge transfer between the fragments can still happen. Therefore, the relevance of such a contribution could be slightly enhanced, even in the covariance analysis.

In order to better understand the origins of these experimental observations, we have calculated photoelectron spectra as a function of carbon–iodine distance while keeping the remaining geometry parameters fixed, for the $^3Q_0$ and $^1Q_1$ excited states (a full description of the theoretical methods is given in "Methods"). As in $CH_3I$[29,52] and other iodoalkanes[29,53–55], photoexcitation occurs predominantly to the $^3Q_0$ state, which correlates to spin–orbit excited $I^*$ $(^2P_{1/2})$ products[56]. This state is crossed by the $^1Q_1$ state, in our case at around 2.4 Å C–I bond distance, correlating to ground state I $(^2P_{3/2})$. From our classical simulations of the C–I bond elongation (described in detail in "Methods"), which reaches an asymptotic velocity of ~25 Å ps$^{-1}$, this channel-crossing occurs at around 10 fs. During the

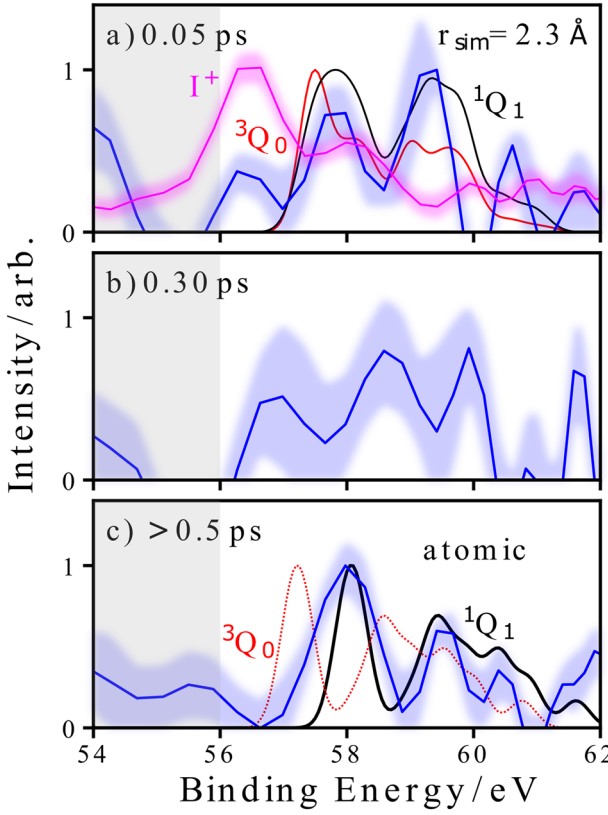

**Fig. 5 Delay-dependent covariant photoelectron spectra.** Angle-integrated electron spectra extracted from the $I^{2+}$ electron–ion partial covariance images at pump-probe delays of 0.05 ps, 0.30 ps, and >0.5 ps, displayed in panels **a**–**c**, respectively, in comparison to selected theoretical results in panels **a**, **c**. For each spectrum, the shaded area in blue represents errors at the 1$\sigma$ level estimated from a bootstrapping analysis. The gray shaded area indicates a level of reduced confidence (see text). In panel **a**, **c**, simulated spectra on the $^3Q_0$ (red) and $^1Q_1$ (black) potentials are shown in comparison to the experimental spectra (blue). In panel **a**, this is for a for C–I bond distance of 2.30 Å, whereas in panel **c**, the theoretical spectra are in the dissociated (i.e., atomic) limits. In panel **a**, the electron spectra extracted from the $I^+$ electron–ion partial covariance image is also shown (in magenta), representing the photoelectron spectrum of unpumped molecules.

dissociation, significant population transfer from $^3Q_0$ to $^1Q_1$ occurs enabling the production of ground-state I atoms. This is the dominant dissociation pathway, particularly in larger alkyl iodides[29,57]. In Fig. 5, the time-resolved experimental data are compared with theoretical spectra calculated close to the equilibrium bond distance (panel a)), and in the long bond distance limit (panel c)) for both electronic states. Our theoretical work does not consider any possible contributions from other excited states, which, in the case of the related $CH_3I$ molecule, are believed to have extremely low oscillator strengths at the pump energy used[58].

Although the time resolution of ± 100 fs precludes direct observation of the non-adiabatic behavior at the conical intersection[57,59,60], the comparison with theory is still illuminating. At the earliest pump-probe delay, potential contributions from the initially populated $^3Q_0$ state and the $^1Q_1$ state cannot be clearly distinguished, with qualitative indications for either state, consistent with some convolution of both involved states. For the second delay at 300 fs displayed in Fig. 5b, the evolution of the spectral dynamics (primarily on the $^1Q_1$ state) is theoretically predicted to be finished, which cannot be decisively confirmed by

the present data. Whether this observation is due to relatively poor statistics for this pump-probe delay, or indicates some longer timescale dynamics (as have been recently observed in $CH_3I$ following A-band excitation[52,61]) cannot be clearly concluded in the current data but will be investigated further in future work. For long delays, in contrast, the dominant contribution can be clearly assigned to the $^1Q_1$ channel ($I(^2P_{3/2})$), with any minor contributions from the $^3Q_0$ state causing a significantly smaller binding energy shift than that observed.

The current work demonstrates the applicability of electron–ion partial covariance imaging to ultrafast site-specific photoelectron spectroscopy. Future upgrades to FELs in terms of repetition rate, polarization[62] and pulse duration[63] control, in combination with ultrashort optical laser pulses in particular in the UV regime[64–66], and advanced camera readout schemes, will enhance the data acquisition rate by orders of magnitude and can hence enable robust determination of covariant electron angular distributions and their temporal evolution. Experiments building on this methodology presented here will enable the study of coupled nuclear electron dynamics, such as those associated with conical intersections, in exquisite detail. Besides gaining first photochemical insights into the chiral molecule 1-iodo-2-methylbutane, our approach can also be readily extended to asymmetric angular distributions, in either the laboratory- or recoil-frame which can provide a promising tool for exploring ultrafast *chiral* dynamics.

## Methods

**Experiment**. Experiments were performed at the beamline BL1 of FLASH1[28], using the CAMP endstation[27]. The experimental setup consists of a dual-sided velocity-map-imaging spectrometer. Samples of R/S-1-iodo-2-methylbutane were introduced as a continuous molecular beam, seeded in helium, which is collimated by two skimmers en route to the spectrometer's interaction region. Here, molecules were crossed perpendicularly by focused UV and XUV pulses provided by the FLASH pump-probe laser and FEL, respectively. Linearly polarized UV pulses (267 nm (4.6 eV), ~150 fs) were generated through the third-harmonic generation of the fundamental output of a Ti:Sapphire amplifier (Coherent Inc., Hidra-25), in a $\beta$-BaB$_2$O$_4$ crystal. The resultant UV pulses were focused in the interaction chamber to a diameter of about 100 μm, with attenuated pulse energies on the order of few μJ (maximum of 7 μJ), and used to photoexcite the target molecules, ultimately initiating C–I bond cleavage.

XUV pulses (19.1 nm (63.5 eV), ~50 fs) generated by FLASH were used to probe the ensuing molecular dynamics through photoionization, predominantly from the I 4d core site[30]. The repetition rate was 10 Hz. The optical laser was blocked by a mechanical chopper for 1 in every 10 pulses (i.e., at 1 Hz), to record background (XUV-only) data. These FEL pulses were circularly polarized using the recently installed four-mirror reflection polarizer in order to potentially enable stereochemical sensitivity[67]. The resultant polarization of the beam is determined by the angle of the mirror assembly, which can be adjusted via a stepper motor. The estimated degree of circular polarization was ~80%. The estimated averaged energy bandwidth of the FEL pulses is approximately 1% full width at half maximum (~0.60 eV), which is the primary contribution to the energy uncertainty in the recorded photoelectron spectra.

The beamline transmission of BL1 at the chosen photon energy is ~18%[27], while the inserted polarizing mirrors have a reflectivity of ~68%. Typical FEL pulse energies were ~50 μJ, but were attenuated by a factor of ~5 using a 420-nm Al filter, in order to reduce contributions from multiphoton effects. Nickel-coated mirrors mounted in Kirkpatrick–Baez geometry focused the beam to about a diameter of about 10 μm.

Following the interaction of target molecules with the focused laser and FEL pulses, the generated ions and electrons are accelerated to position-sensitive MCP/ phosphor screen detectors at the top and bottom of the instrument, respectively. Potentials were applied to the ion optics such that velocity mapping conditions were met for both ions and electrons[31]. On the ion side, the resultant light flashes at the phosphor are imaged by a fast-time-stamping Pixel Imaging Mass Spectrometry (PImMS) camera[32,33]. This employs a 324 × 324 pixel sensor capable of recording the spatial coordinates (x, y) and arrival time (t) of events at high count rates. In the current experiment, the sensor was operated at a timing precision of 25 ns, which facilitates imaging of a wide range of ions within a single experimental cycle. Velocity-map images corresponding to a particular m/z value were extracted from the PImMS dataset by integrating over a characteristic time-of-flight range for the ion of interest. Multi-mass imaging allows momentum correlations between ionic fragments to be determined using covariance analysis[20,21,39]. The electron detector was gated in time by fast HV switches

(Behlke) to minimize background contributions from stray light, and the electron images were captured using a 1388 × 1038 pixel CCD camera.

Ion and electron velocity-map images were recorded for several pump-probe delays between the optical and free-electron lasers, as were relevant single-color and background datasets. The fluctuations of the FEL timing and pulse energy on a shot-to-shot basis were recorded using the FLASH Bunch Arrival Monitor[49] and Gas Monitor Detector[68], respectively. Throughout the beamtime, data at several fixed pump-probe delays were recorded by switching delays between individual (~1–2 h) data acquisition runs, to minimize the effect of gradual drifts in experimental conditions on the data at a given pump-probe delay. Frequently, acquisitions were also recorded whilst scanning the pump-probe delay in small steps, in order to conclusively establish time zero and thus verify stable timing between the optical and FEL pulses. In order to improve the three-dimensional (x, y, t) resolution of the ion-imaging data recorded by the PImMS camera, centroiding in time and space was performed[39].

**Theoretical methods**. Theoretically, the cross-section for ionizing from spin–orbit (SO) coupled states I to SO-coupled state F were calculated as

$$\sigma_{IF}(\omega) = \frac{4}{3}\alpha\pi^2\omega\sum_{I'}\sum_{F'}|C_{I',I}|^2|C_{F',F}|^2\sum_{M=-1,0,1}\left|\left\langle\psi_{F'}^N|\hat{d}_M|\psi_{I'}^N\right\rangle\right|^2\delta(E_F - E_I - \omega) \quad (1)$$

where $\alpha$ is the fine structure constant, $C_{I',I}$ and $C_{F',F}$ are the expansion coefficients of the initial and final SO-coupled states in the non-SO-coupled basis states $|\psi_{I'}^N\rangle$ and $|\psi_{F'}^N\rangle$, respectively, and $\delta(E_F - E_I - \omega)$ is a Gaussian broadening function with a standard deviation of 0.212 eV. The many electron dipole operator in Eq. (1) is

$$\left\langle\psi_{F'}^N|\hat{d}_M|\psi_{I'}^N\right\rangle = \sum_{i,j}\left\langle\psi_{F'}^N|c_i^\dagger c_j|\psi_{I'}^N\right\rangle\langle i|\hat{d}_M|j\rangle, \quad (2)$$

where $c_i$ and $c_j^\dagger$ are fermionic creation and annihilation operators and $\langle i|\hat{d}_M|j\rangle$ the dipole moment in the one-particle basis.

To calculate the transition dipole matrix elements for the basis states, $\left|\left\langle\psi_{F'}^N|\hat{d}_M|\psi_{I'}^N\right\rangle\right|^2$, we employed the one-center approximation, in which we separated the final electronic state $|\psi_F^N\rangle$ into a bound part $|\psi_F^{(N-1)}\rangle$ and a one electron continuum part $\phi_k$. The respective continuum wave function $\phi_k$ was approximated with atomic continuum wave functions at the appropriate kinetic energy $k^2/2 = \omega - (E_F - E_I)$. More details on this procedure can be found in refs. [34,69].

We obtained the expansion coefficients $C_{I',I}$ and $C_{F',F}$, as well as $E_I$ and $E_F$ from SO calculations diagonalizing the Breit–Pauli Hamiltonian in the space of the initial and final basis states $\psi_{I'}^N$ and $\psi_{F'}^{(N-1)}$, respectively. In particular, we conducted a state-averaged complete active space (SA-CASSCF) calculation involving an orbital space of four orbitals with six electrons for an equal-weighted average over the three lowest singlet states employing the 6-311G(d,p) basis set[70,71]. With the obtained set of orbitals, we constructed a set of basis states for the SO calculations for the initial, neutral state consisting of the six lowest triplet and four lowest singlet states obtained by diagonalizing the configuration interaction (CI) matrix in this active space. For the final state, we took into account the full spectrum of states constructed by diagonalizing the CI matrix in the employed active space with an additional hole in the 4d shell resulting in 80 doublet and 30 quadruplet states. The SO coefficients $C_{I,I'}$ and $C_{F,F'}$ as well as the respective energetic position of the SO-coupled states were calculated using molpro version 2020.1[72]. The cross-sections were calculated using the XMOLECULE toolkit[69,73].

To calculate the molecular photoelectron spectrum, we employed a common orbital set for the initial, neutral, and final 4d-ionized states. Because orbital relaxation effects due to the presence of the 4d hole are not taken into account, the energy differences between initial and final states are somewhat too large. This effect has been estimated from a calculation for atomic iodine where the orbital relaxation effect was considered. We find that the inclusion of relaxation effects results in a spectral shift of ~5 eV and only minor changes in the spectral shape. We further note that discrepancies to the experimental spectra may also arise due to missing relativistic effects and the limited size of the employed basis set. To correct for these effects and facilitate a comparison with the experimental data, we additionally shifted the calculated photoelectron binding energies by 1.4 eV towards lower energies.

Figure 6 shows the calculated spectrum for the molecular electronic ground-state. As can be seen, the calculation shows the expected SO splitting of ~1.7 eV between the two levels 4d$_{5/2}$ and 4d$_{3/2}$ levels[74].

For the excited state, we have calculated the photoelectron spectrum as a function of iodine carbon distance, keeping the remaining geometry parameters fixed. Specifically, we show spectra, for the excited $^3Q_0$ and $^1Q_1$ state, that are relevant for the photoinduced dissociation dynamics. Figure 7 shows the calculated potential energy curves. The $^3Q_0$ state correlates asymptotically to the excited iodine fragment in $^2P_{1/2}$ configuration (I*), the $^1Q_1$ state corresponds to the iodine in its $^2P_{3/2}$ ground state. As can be seen, both potential energy curves cross at ~2.4 Å.

The calculated 4d photoelectron spectrum as a function of internuclear distance is shown in Fig. 8 for the $^3Q_0$ state and in Fig. 9 for the $^1Q_1$ state. With increasing interatomic distance, one can see that both spectra initially move to lower binding

energies and exhibit only slight changes beyond an internuclear C–I distance of 3.5 Å. The $^3Q_0$ photoelectron spectrum at large internuclear distances is roughly 0.9 eV lower compared to the $^1Q_1$ photoelectron spectrum. As expected from the asymptotic dissociation limit, the $^3Q_0$ spectrum corresponds to the atomic I* photoelectron spectrum at large internuclear distances, whereas the $^1Q_1$ spectrum corresponds to the atomic I ground-state photoelectron spectrum.

To qualitatively assess the dynamics triggered by the initial excitation, we conducted an MD simulation. Starting from the Franck–Condon geometry of the molecule, the trajectories were propagated on the lowest triplet excited state using the TD-DFT method employing the SBKJC effective-core potential basis set[75] using GAMESS[76]. As we described in the main text, we observe that the C–I bond distance almost linearly increases with a speed of $\simeq 25$ Å ps$^{-1}$ in accordance with earlier results for similar iodoalkanes[29]. We see that the conical intersection at 2.4 Å is reached within $\simeq 10$ fs, as displayed in Fig. 10.

**Assignment of Time Zero**. In experiments incorporating a weak-field UV pump and an XUV probe, it is often difficult to precisely assign time zero[4,7,8,77]. For instance, analysis of the low KE, neutral photodissociation feature observed in the $I^{2+}$ ion signal in the current work is complicated by the fact that the feature's time evolution depends on delay-dependent (and therefore distance-dependent) charge-transfer probabilities following photoexcitation. As a result, in the present experiment, time zero is determined from the delay-dependent intensity of the higher KE feature observed in the $I^{2+}$ ion signal, whose KE decreases at longer pump-probe delays. As mentioned in the main text, this feature arises from a multiphoton UV dissociative ionization to yield a cationic alkyl fragment, before XUV ionization at the neutral iodine fragment. As we expect this channel to be observed immediately following UV excitation, and charge transfer is much less favorable to an already charged alkyl fragment (and thus less likely to affect the yield of this channel), we take the center of the rise of this channel as the point at

which the two pulses are temporally overlapped. The intensity of this feature as a function of pump-probe delay, and a fit to this, from which the time zero may be extracted, is shown in Fig. 11.

**Error estimation**. Figure 4 of the main manuscript displays the delay-dependent electron spectra in partial covariance with the $I^{2+}$ ion. Estimated errors of these spectra ($1\sigma$) are also given, represented as shaded areas. These errors were determined using a bootstrapping analysis of the data. The data used to generate each spectrum was originally recorded as multiple distinct data acquisitions (each of tens of thousands of laser shots) during the original FEL beamtime. For each pump-probe delay, these individual data acquisitions were randomly sampled (with replacement), to generate a new dataset, from which the electron–ion partial covariances were calculated. This process was repeated many times, and the standard deviation from the many spectra was used as an estimate of the overall statistical error.

**Electron–ion partial covariance calculation**. Covariance, a measure of linear correlation between two variables, $X$ and $Y$, is defined as[20]:

$$\text{Cov}(X, Y) = \langle XY \rangle - \langle X \rangle \langle Y \rangle \qquad (3)$$

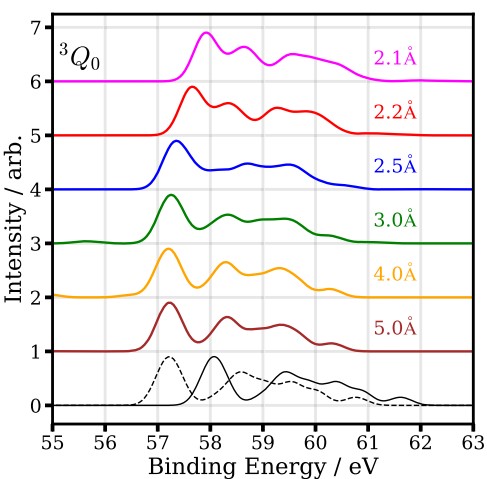

**Fig. 8 Calculated photoelectron spectrum of the 4d level of the molecule ($^3Q_0$ excited state) for selected interatomic distances.** The two lowest lines show the calculated spectra for atomic iodine in its ground (black, solid line) and excited state (black, dotted line).

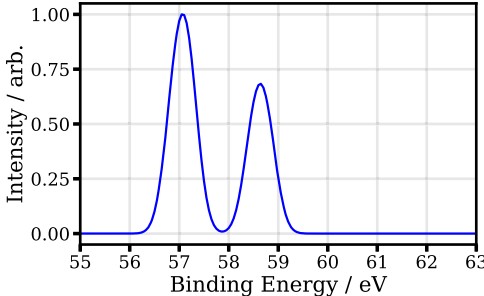

**Fig. 6 Calculated photoelectron spectrum of the 4d level of 1-iodo-2-methylbutane.** Ground-state binding energies and normalized yields of the investigated 4d electrons of 1-iodo-2-methylbutane are depicted in the relevant energy range.

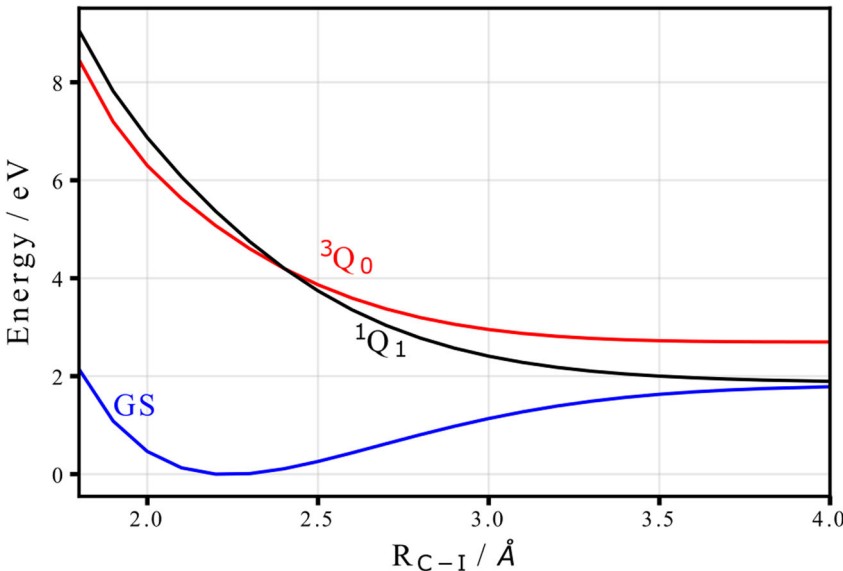

**Fig. 7 Calculated potential energy curves for the ground and selected excited states of the molecule along the C–I bond.** The main contributing states to the findings, i.e., the ground state as well as the $^3Q_0$, and $^1Q_1$ excited states, are depicted in blue, red and black, respectively.

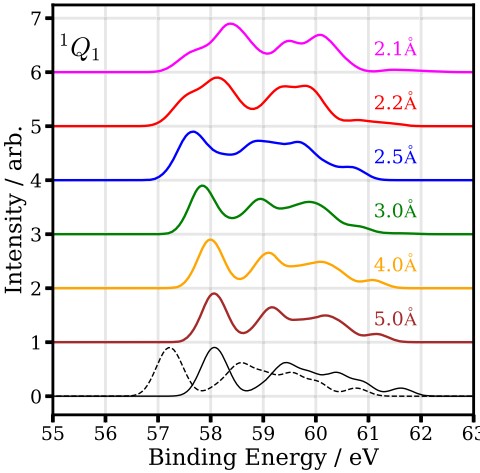

**Fig. 9 Calculated photoelectron spectrum of the 4d level of the molecule ($^1Q_1$ excited state) for selected interatomic distances.** The two lowest lines show the calculated spectra for atomic iodine in its ground state (black, solid line) and excited state (black, dotted line).

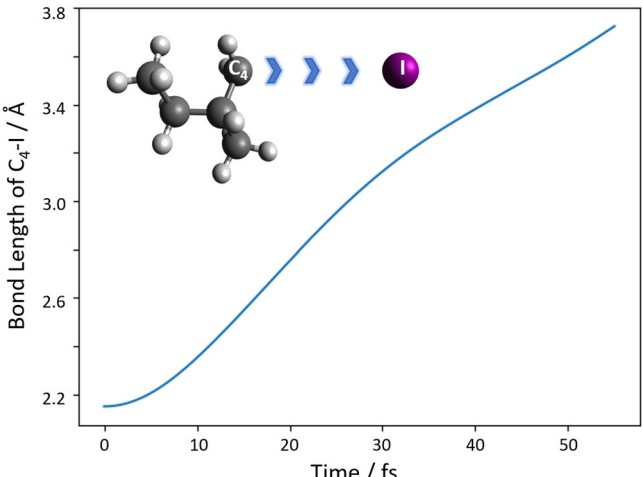

**Fig. 10 Simulation of the bond-length elongation between the closest carbon ($C_4$) and the iodine atom.** The inset illustrates the dissociation process which is quantitatively depicted for the calculations between 0 fs and 50 fs.

where arithmetic means are taken over a series of observations (in our case, laser shots). In this work, prior to the calculation of the covariance, some data filtering was performed. First, only laser shots in which both the UV and XUV lasers irradiated the sample were selected. Second, FEL pulses with outlying pulse energies (more than $1.5\sigma$ away from the mean pulse energy) were removed. The covariance was then calculated between each pixel of the electron image (recorded on a shot-to-shot basis) and the total count of the ion channel in interest, derived from the centroided PImMS camera data. Such a covariance image is shown in the left-hand side of Fig. 2b of the main manuscript.

The calculated covariance images still feature signal which is not correlated to the ion of interest. This "false covariance" arises due to the fluctuating FEL power during the experiment, as described in the main text. To account for this, partial covariance is calculated[25], in which an additional correction term, representing these (linear) correlations induced by the fluctuating power, is calculated. This term is defined as:

$$\mathrm{Corr}(X, Y; I) = \frac{\mathrm{Cov}(Y, I)\mathrm{Cov}(I, X)}{\mathrm{Cov}(I, I)} \qquad (4)$$

The correction term is calculated using the FEL pulse energy measured by the FLASH Gas Monitor Detector[49] as the fluctuating parameter. The correction term is calculated by determining the covariance between the FEL pulse energy and the

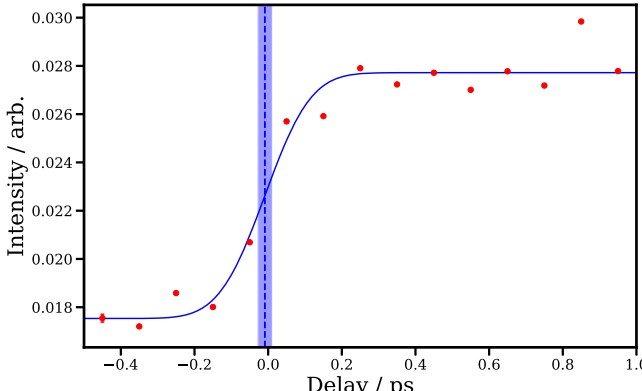

**Fig. 11 Pump-probe delay-dependent intensity (red points) of the $I^{2+}$ "Coulomb curve" feature.** A Gaussian cumulative distribution function (CDF) fit to this is shown in blue. The center of this fit is marked by a blue dashed line, and the standard fitting error of this parameter is marked by the shaded blue region.

ion count, as well as the covariance between the FEL pulse energy and each pixel of the electron image, as seen in Eq. (4). Once the correction term is subtracted from the covariance, the partial covariance remains. Example correction and partial covariance images are shown in the center and right-hand portions of Fig. 2b in the main manuscript. The covariant electron spectra presented in the main manuscript are obtained by Abel-inverting the electron–ion partial covariance images (using the pBASEX[78] algorithm). The presented spectra are normalized to unit peak intensity in the spectral region of interest.

## Data availability

The data that support the findings of this study are available from the corresponding author on reasonable request. All steps to reproduce the presented experimental and theoretical findings are either explained in detail or cited in the manuscript.

## Code availability

The codes used to process and analyze the experimental data, including to calculate the electron–ion partial covariances, are available from the corresponding authors on reasonable request.

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

## Acknowledgements

We gratefully acknowledge DESY for the provision of the free-electron laser beamtime at the BL1 CAMP endstation of FLASH as well as the work and sedulous support of the scientific and technical teams. Beamtime was allocated for proposal F20171078. F.A., M.B., D.H., R.M., and C.V. acknowledge funding from EPSRC Programme Grants EP/L005913/1 and EP/T021675/1. M.Bu. is grateful for support from the UK EPSRC (EP/S028617/1 and EP/L005913/1). This work was partly funded by the Deutsche Forschungsgemeinschaft (DFG)-Project No. 328961117-SFB 1319 ELCH (Extreme light for sensing and driving molecular chirality). We furthermore acknowledge the Max Planck Society for funding the development and the initial operation of the CAMP endstation within the Max Planck Advanced Study Group at CFEL and for providing this equipment for CAMP@FLASH. The installation of CAMP@FLASH was partially funded by the BMBF grants 05K10KT2, 05K13KT2, 05K16KT3, and 05K10KTB from FSP-302. M.I., V.M., and Ph.S. acknowledge funding by the Volkswagen Foundation within a Peter-Paul-Ewald Fellowship. S.D., K.S., L.S., and S.B. acknowledge funding from the Helmholtz Initiative and Networking Fund through the Young Investigators Group Program (VH-NG-1104). Furthermore, K.S. and S.B. were supported by the Deutsche Forschungsgemeinschaft, project B03 in the SFB 755 - Nanoscale Photonic Imaging. D.Ro. was supported by the US National Science Foundation through grant PHYS-1753324.

## Author contributions

M.I. conceived and proposed the experiment. The experiment was performed by F.A., V.M., R.B., B.E., Ph.S., T.M.B., M.Bu., S.D., P.G., D.H., M.L., J.W.L.L., L.M., R.M., H.O., C.P., D.Ro., K.S., L.S., R.W., V.Z., S.B., and M.I. The optical laser system was operated and adjusted by B.M., R.B., B.E., and P.G. with support in the design by A.G. Engineering support including modifications to the spectrometer were provided by B.E. and D.R. The R-enantiomer of the target was synthesized and provided by R.P., D.K., and I.V. F.A. analyzed the data with further contributions during the beamtime from Ph.S., L.M., C.P., and S.D. The electron–ion partial covariance methods were developed by F.A. with insight from M.B. and M.I. L.I. and Z.L. performed the theoretical calculations. Cv.K.S. has set up and supported the operation of the polarizing mirrors. F.A., V.M., L.I., R.B., B.E., Ph.S., T.M.B., G.B., M.Bu., Ph.D., S.D., A.E., A.G., P.G., D.H., D.K., M.L., J.W.L.L., Z.L., B.M., L.M., R.M., M.M., H.O., C.P., R.P., D.R., D.Ro., K.S., L.S., R.D.T., C.V., I.V., Cv.K.S., R.W., P.W., V.Z., S.B., M.B., and M.I. contributed to in-depth discussions and interpretation. F.A., L.I., and M.I. wrote the manuscript with dedicated contributions by Z.L.

## Funding

## Competing interests

The authors declare no competing interests.
