## [Peer Review File · Communications Chemistry]

Reviewers' comments:

Reviewer #1 (Remarks to the Author):

It is an interesting paper showing the abilities of electron-ion partial covariance to isolate photoelectron spectrum arising from different photofragmentation pathways. The present technique in FEL could be interesting for communities of studying structural and chemical changes from a specific spectator site. Generally, the paper looks fine, it presents the experimental and theoretical results in a normal way, but the present version does not touch me too much, but confuse me with the experimental complex techniques. Maybe the most important information should be highlighted for the broad readers. Other specific comments: (1) Caption of Figure 1 and related text are too simple to understand its full meaning; (2) Fig.2a, what is the time delay for the ion mass spectra for the 'blue' one?; (3) Fig3 and related text, how to full subtract the back ground of He is not clearly presented; (4) first paragraph of p14, 'Figure 4' should be 'Figure 5'?; (5) Figure 8, '(black, solid line' and '(black, dotted line)' should interchange?.

Reviewer #2 (Remarks to the Author):

This paper studied the ultraviolet photodissociation of 1-iodo-2-methylbutane, probed by XUV pulses from the FLASH facility. The electron-ion partial covariance imaging technique was used to isolate otherwise elusive features in a two-dimensional photoelectron spectrum arising from different photofragmentation pathways. This method will enable the study of coupled nuclear electron dynamics, such as those associated with conical intersections. The result is interesting and it is deserve to be published in Communications Chemistry.

Some issues to address below:

1. Is the UV pulse linearly or circularly polarized? The two places of description about the polarization seems to be contradictory. "...is peaked along the UV polarization axis...", "These FEL pulses were circularly polarized using...".
2. The velocity-map I^{2+} ion images shown in Fig.2 are obviously depends on the angular direction of detection, but the coordinate system is not given here.
3. The physical means of the notations β and β_2 should be given explicitly.
4. There are places of incomplete informations throughout the references.

Reviewer #3 (Remarks to the Author):

This is an interesting and very well-written paper, reporting on the use of electron-ion partial covariance imaging to study ultrafast dissociation dynamics. The authors present state-of-the-art time-resolved UV-XUV pump-probe experiments making use of the FLASH free electron laser. In order to unravel the dissociation dynamics they apply for the first time a partial covariance technique to extract time-dependent photoelectron spectra correlated to a specific structural site in the dissociating molecule. They successfully demonstrate the applicability of the method on a rather complex molecule, 1-iodo-2-methylbutane. And even though the statistics on the experiments could have been improved and some questions remain as to the interpretation of the data, this work represents an important pathfinder study with many interesting potential applications related to

ultrafast chemistry. As such, I think that the paper is certainly suited for publication in Communications Chemistry.

However, I would like to raise a few points that the authors may want to consider.

1. I am surprised as to why the authors chose such a rather complex molecule for their first demonstration of the partial covariance method applied to electron-ion imaging. Of course it is very interesting to demonstrate that the method still works for a complex molecule, but would a simpler molecule like CH₃I, for which ample previous data already exists, not have been better suited? It also has a conical intersection (of the 3Q0 and 1Q1 states) and I would expect that the time-dependent photo-electron spectra would show similar shifts.

2. The authors discuss their error estimation using bootstrapping methods, leading to the errors in the PES spectra shown in Figure 5. However, it is not clear what the errors and dominant contributions in the binding energy are. What is the FEL bandwidth, what the resolution obtained in the VMI images, and how does it compare to the small shifts analysed in the time-resolved PES spectra? Could there be a jitter between measurements that could explain the somewhat unexplained evolution of PES signal shown in Fig. 5? I find this in particular important to discuss in view of the large shift of 6.4 eV subtracted from the calculated binding energies and the apparent low energy feature (below 57 eV) in Figures 5 a) and c).

3. The authors show in Figure 2 c) a fit to a CDF of the integrated yield of the low energy I⁺ signal over time and give a sigma value, but this is not further discussed in the text. Could this analysis be used to learn more about the apparent charge transfer processes as a function of internuclear distance? And how does this behaviour influence the interpretation of the PES spectra at small delay times shown in Figure 5?

4. To me it seems that in Figure 3 c) the noise level in the partial covariance electron spectrum is higher than in the BG subtracted velocity map image. Could the authors mention why this is the case or if the observed structure at higher binding energies might correspond to real features, as one might think from the comparison to theoretical values in Figure 5?

5. I was wondering why He (and not Ne, or Ar) was used as a carrier gas in the experiments, although this leads to background signal close to the expected iodide signal.

6. In figure 5 c), it looks as if the calculated 3Q0 and 1Q1 binding energy curves are shifted compared to Figure 8 and 9. Note: in Figure 8 the assignment for atomic iodine is wrong (excited, ground state interchanged). I was also wondering if any other excited states were considered, and if these could help to explain the current mismatch with the experimental spectrum at 300 fs (Fig. 5 b). For the interpretation of Figure 5 it might also have been nice to show a spectrum corresponding to UV off, i.e., corresponding to the ground state spectrum of the molecule (Figure 6). I assume that the I⁺ signal was too low for the partial covariance analysis in the absence of the XUV pulse?

List of answers and changes COMMSCHEM-21-0356-T

Reviewer #1 (Remarks to the Author):

It is an interesting paper showing the abilities of electron-ion partial covariance to isolate photoelectron spectrum arising from different photofragmentation pathways. The present technique in FEL could be interesting for communities of studying structural and chemical changes from a specific spectator site. Generally, the paper looks fine, it presents the experimental and theoretical results in a normal way, but the present version does not touch me too much, but confuse me with the experimental complex techniques. Maybe the most important information should be highlighted for the broad readers.

We thank the reviewer for their positive feedback, and for raising the general point about highlighting the most important information for a broader readership. To emphasize this in the revised manuscript we have implemented the following changes in order to better guide the reader through the individual findings and argumentations:

Abstract (also in the light of Rev. #3 asking for clarification of the molecular choice):

“...of the prototypical chiral molecule...”

Middle of the first technical section “Velocity-map Ion Imaging” we have now elaborated on the angular distribution in order to allow accessing the relevance of this information also by non-specialists:

“As is expected for one-photon transitions, intensity of this feature as a function of angle to the polarization axis, $I(\theta)$, is of the form $I(\theta) = (\sigma/4\pi)[1 + \beta P_2 \cos\theta]$, where P_2 is the second Legendre polynomial, and β is the anisotropy parameter. This takes limiting values of -1 and +2 for transitions of pure perpendicular or parallel nature, respectively (under the assumption of a prompt photodissociation) [36,37] In the present case, a β value of ≈ 1.80 is extracted.”

End of the first technical section “Velocity-map Ion Imaging”:

“As shown in this section, the temporally and kinetic-energy resolved ion-yield evolution already provides valuable information about the individual dissociation channels and allows to partially disentangle them. Deeper insights about selective contributions and processes can then be accessed by studying the underlying electronic dynamics.”

For the relevance of the electron-ion covariance findings, we refer to the paragraph on top of page 12 that is meant to guide the reader towards the next and final step.

“Crucially, and in contrast to previous work, this energetic shift as a result of dissociation can be completely isolated from the far stronger unpumped parent molecule signal, as well as from any competing pump-probe channels, such as the multiphoton dissociative ionization pathway. As such, the method presented here allows for decisive insights into the photochemistry of this prototypical molecule.”

The required technical and methodological explanations before this are, to our feeling, necessary for concluding the interpretation and for accessing the relevance of the following main results that are then summarized and put in perspective to future developments before the Methods section.

We agree that our work is presented in a way that the methodological advances and the scientific results are equally weighted in their importance and that this can be experienced in a way as the reviewer describes. Besides the above sketched changes and explanations, we, however, feel that the content and structure of the manuscript would need to be much more substantially altered in order to specifically highlight certain aspects over others, which, also in the light of the otherwise positive reviews, seems too impactful to our feeling.

Finally, we note that in addressing the following specific comment, a more general overview of the experimental methodology is given in a way that is useful to a broad readership.

Other specific comments:

(1) Caption of Figure 1 and related text are too simple to understand its full meaning;

We have elaborated on the text in this caption and the text surrounding it, which gives an overview of the time-resolved inner-shell photoelectron spectroscopy approach. The caption now reads:

“Schematic representation of the experimental scheme to study the ultrafast photodynamics of 1-iodo-2-methylbutane. Photoexcitation (predominantly to the excited state of 3Q_0 symmetry) is initiated by a UV pump pulse. The photoexcited molecule is interrogated at a series of pump-probe delays by a XUV-FEL pulse, probing the photoelectron binding energy of the I (4d) core orbital. Measured changes in the binding energy during the photodissociation can be related to the underlying photochemistry, supported by quantum simulations of the photoionization process.”

(2) Fig.2a, what is the time delay for the ion mass spectra for the 'blue' one?;

The example mass spectrum is acquired over a delay of +0.80ps. We have added a comment in the caption of Figure 2 mentioning this:

“Normalized ion mass spectra recorded for 1-iodo-2-methylbutane with the XUVonly (magenta), UV only (red) and UV early-XUV late at a pump-probe delay of +0.80 ps (blue).”

(3) Fig3 and related text, how to full subtract the background of He is not clearly presented;

Here, the Helium only images and Helium+molecule images were normalized by the number of laser shots in the respective acquisitions and the average FEL pulse energy. The two normalized images are then subtracted from one another. This is now clarified in the text:

“Subtraction of the helium-only background image, normalized by number of laser shots and average FEL pulse energy yields the image plotted on the right of panel a) of Figure 3”

(4) first paragraph of p14, 'Figure 4' should be 'Figure 5'?

We have corrected this typo.

(5) Figure 8, '(black, solid line' and '(black, dotted line)' should interchange?.

We thank the reviewer for identifying this error in the caption of Figure 8. We have corrected this error.

Reviewer #2 (Remarks to the Author):

This paper studied the ultraviolet photodissociation of 1-iodo-2-methylbutane, probed by XUV pulses from the FLASH facility. The electron-ion partial covariance imaging technique was used to isolate otherwise elusive features in a two-dimensional photoelectron spectrum arising from different photofragmentation pathways. This method will enable the study of coupled nuclear electron dynamics, such as those associated with conical intersections. The result is interesting and it is deserve to be published in Communications Chemistry.

We thank the reviewer for their positive assessment of the manuscript, and their detailed feedback.

Some issues to address below:

1. Is the UV pulse linearly or circularly polarized? The two places of description about the polarization seems to be contradictory. "...is peaked along the UV polarization axis..."; "These FEL pulses were circularly polarized using...".

We thank the reviewer for spotting a lack of clarity on this point. The UV pulses are linearly polarized, whilst the XUV pulses are circularly polarized. We have altered the following sentence in the methods section:

"Linearly polarized UV pulses (267 nm (4.6 eV), ~150 fs) were generated through third-harmonic generation of the fundamental output of a Ti:Sapphire amplifier (Coherent Inc., Hydra), in a β -BaB₂O₄ crystal."

In addressing the following point we have also clarified this further.

2. The velocity-map I^{2+} ion images shown in Fig.2 are obviously depends on the angular direction of detection, but the coordinate system is not given here.

The angular coordinate system in the VMI images in Figure 2 is defined by the polarization axis of the UV laser, as is indicated by the vertical white arrow. We have added the following text to the caption of Figure 2 to clarify this:

"The polarization axis of the UV laser is vertical in these images."

3. The physical means of the notations β and β_2 should be given explicitly.

We have added the following text when discussing the angular distribution of the ion image:

“As is expected for one-photon transitions, intensity of this feature as a function of angle to the polarization axis, $I(\theta)$, is of the form $I(\theta) = (\sigma/4\pi)[1 + \beta P_2(\cos\theta)]$ where P_2 is the second Legendre polynomial, and β is the anisotropy parameter. This takes limiting values of -1 and +2 for transitions of pure perpendicular or parallel nature, respectively (under the assumption of a prompt photodissociation). In the present case, a β value of ≈ 1.80 is extracted.”

We have also included an additional reference to the Angular Momentum textbook by Zare, which covers this topic extensively.

4. There are places of incomplete informations throughout the references.

We thank the reviewer for spotting this. We have added missing information to several references.

Reviewer #3 (Remarks to the Author):

This is an interesting and very well-written paper, reporting on the use of electron-ion partial covariance imaging to study ultrafast dissociation dynamics. The authors present state-of-the-art time-resolved UV-XUV pump-probe experiments making use of the FLASH free electron laser. In order to unravel the dissociation dynamics they apply for the first time a partial covariance technique to extract time-dependent photoelectron spectra correlated to a specific structural site in the dissociating molecule. They successfully demonstrate the applicability of the method on a rather complex molecule, 1-iodo-2-methylbutane. And even though the statistics on the experiments could have been improved and some questions remain as to the interpretation of the data, this work represents an important pathfinder study with many interesting potential applications related to ultrafast chemistry. As such, I think that the paper is certainly suited for publication in Communications Chemistry.

We thank the reviewer for their kind assessment of the manuscript.

However, I would like to raise a few points that the authors may want to consider.

1. I am surprised as to why the authors chose such a rather complex molecule for their first demonstration of the partial covariance method applied to electron-ion imaging. Of course it is very interesting to demonstrate that the method still works for a complex molecule, but would a simpler molecule like CH₃I, for which ample previous data already exists, not have been better suited? It also has a conical intersection (of the 3Q0 and 1Q1 states) and I would expect that the time-dependent photo-electron spectra would show similar shifts.

We absolutely agree that CH₃I would have been the easier choice for specifically this kind of study, but the current experiment constitutes a first step towards using circularly polarized FEL light to probe ultrafast chiral dynamics (through photoelectron circular dichroism). The presented benchmarks and findings for iodomethylbutane are the basis to enable FELs to site-selectively investigate individual electronic channels in chiral systems and evolving fragments in a next step. Although the present works lays the foundation for these kinds of studies, we refrained from stressing this background too prominently since it may have made the wrong impression of topical orientation and resulting claims of this manuscript. To take the comment into account and make the choice clearer from the beginning on, we have added the following text to the abstract:

“Here, we study the ultraviolet photodissociation of the prototypical chiral molecule 1-iodo-2-methylbutane, probed by XUV pulses from the Free-electron LASer in Hamburg (FLASH) through the ultrafast evolution of the iodine 4d binding energy.”

We also kindly refer to the passages where more on this background is elaborated on in lines 39-43 and 247-250.

2. The authors discuss their error estimation using bootstrapping methods, leading to the errors in the PES spectra shown in Figure 5. However, it is not clear what the errors and dominant contributions in the binding energy are. What is the FEL bandwidth, what the resolution obtained in the VMI images, and how does it compare to the small shifts analyzed in the time-resolved PES spectra? Could there be a jitter between measurements that could explain the somewhat unexplained evolution of PES signal shown in Fig. 5? I find this in particular important to discuss in view of the large shift of 6.4 eV subtracted from the calculated binding energies and the apparent low energy feature (below 57 eV) in Figures 5 a) and c).

The estimated average energy bandwidth of the FEL pulse is ~1% (~0.6 eV) FWHM, whereas the estimated KE resolution of the electron spectrometer is significantly better than this. During the beamtime, the pump-probe delay was changed between individual runs (typically each of 1-2 hour acquisition time). As such, drifting experimental conditions over time would not be expected to affect one pump-probe delay more often than others. As well as recording data at the static delays analyzed in the manuscript, shorter scans over a range of pump-probe delay scans were also recorded regularly. This allowed the quick identification of and correction for any drifts in laser-FEL timing. We have not identified any drifts or possible changes in experimental conditions which would lead to the (as the reviewer correctly says, as yet unexplained) PES signal at 300 fs pump-probe delay. We have elaborated on these points in the experimental methods section:

“The estimated averaged energy bandwidth of the FEL pulses is approximately 1% FWHM (~0.60 eV), which is the primary contribution to the energy uncertainty in the recorded photoelectron spectra.”

and

“Throughout the beamtime, data at several fixed pump-probe delays were recorded by switching delays between individual (~1-2 hour) data acquisition runs, to minimize the effect of gradual drifts in experimental conditions on the data at a given pump-probe delay. Frequently, acquisitions were also recorded whilst scanning the pump-probe delay in small steps, in order to conclusively establish time-zero and thus verify stable timing between the optical and FEL pulses.”

The shift of 6.4 eV that is applied in the calculation includes mainly the missing relaxation contributions in the calculation. Such a shift is not untypical considering the

fact that core excitation involves some strong orbital relaxation contributions that are in practice difficult to include in the molecular calculation. Because we can clearly confirm the extent and the direction of this energy shift by comparison of atomic iodine calculations, we are confident that the shift of ~ 6.4 eV yields a realistic spectrum.

3. The authors show in Figure 2 c) a fit to a CDF of the integrated yield of the low energy I^{2+} signal over time and give a sigma value, but this is not further discussed in the text. Could this analysis be used to learn more about the apparent charge transfer processes as a function of internuclear distance? And how does this behaviour influence the interpretation of the PES spectra at small delay times shown in Figure 5?

We thank the reviewer for raising this point which was not clearly discussed in the main text. The possibility of charge-transfer at short internuclear separations/pump-probe delays is expected to cause a temporal shift in the rise of this feature. Such phenomena has now been studied in several FEL experiments, references to which have now been added in the revised manuscript. Typically, the delay-dependent intensity of the low KER ions is analyzed for a range of ion charge states, and shifts between different charge states provides information about how distance-dependent charge-transfer probabilities vary with charge state. However, in the present work, which uses relatively weak XUV pulses (to maintain in a predominantly 1 photon ionization regime) and a photon energy only a few eV above the iodine 4d edge, we do not generate a range of iodine charge states (which require multiple photon absorption). This greatly limits the insight into charge transfer which can be extracted in the current experiment. The PES spectra shown in Figure 5 are extracted through electron-ion covariance for the low KE ions (which are formed if charge transfer does not occur). We note that future work which could (with much higher resolution) probe the photoelectron (and Auger electron) spectra during the times over which charge transfer can occur would be very interesting, and would likely benefit from the electron-ion partial covariance analysis procedure introduced in the current work.

We have added the following text to the manuscript during the discussion of Figure 2 elaborating on these points:

“This signal rises on an ultrafast (few hundred fs) timescale, as is expected for a direct dissociation, as observed previously in related alkyl iodides. The rise in intensity of this feature is somewhat delayed with respect to time-zero. This is to be expected as, at sufficiently early pump-probe delays, charge transfer can occur between the multiply charged iodide ion produced following XUV ionization and the recoiling C_4H_9 radical, reducing the low energy I^{2+} ions formed. Previous pump-probe studies using site-selective ionization in similar photodissociation molecules have examined differences in

the delay-dependent behavior of multiple iodine charge states to extract information about distance-dependent charge-transfer probabilities. However, in the present work, which employs a relatively weak XUV pulse which is only a few eV above the I 4d binding energy of the neutral molecule, a range of iodine ion charge states are not populated, and thus the extractable insights into charge transfer are limited and not discussed further in the present manuscript.”

To further elaborate on the observed and previously undiscussed feature at 56.5 eV for low delay (see Fig. 5a), we added the following text from lines 204 to 208:

“The peak at ~56.5 eV visible in panel a) coincides energetically with the signal stemming from the unpumped molecule, indicated by the I⁺ signal and displayed in more detail in Figure 3 c). At small time delays, the overall yield of the I²⁺ is reduced since charge transfer between the fragments can still happen. Therefore, the relevance of such a contribution could be slightly enhanced, even in the covariance analysis.”

4. To me it seems that in Figure 3 c) the noise level in the partial covariance electron spectrum is higher than in the BG subtracted velocity map image. Could the authors mention why this is the case or if the observed structure at higher binding energies might correspond to real features, as one might think from the comparison to theoretical values in Figure 5?

With regards to the increased noise in the electron-ion partial covariance maps, this is true. There are several reasons for this:

- Inherently, covariance analysis results in greater noise than is observed in the equivalent uncorrelated data as recorded. This has been discussed theoretically in previous works, and is compounded by finite detection efficiencies in a real experiment. In the current example, the amount of correlated signal is reduced by the product of detection efficiency of the ion of interest and the electron.
- In the case shown in panel c), in the covariance case, the spectrum plotted is only for electrons in covariance with the I⁺ ion. Selecting a single ion in this manner necessarily increases the relative contribution of noise. This is very strongly observed in the electron images in correlation with the I²⁺ ion.

We have added the following text to the manuscript discussing the noise in the covariant electron spectra, which also now cites two papers which generally examine the factors influencing in covariance experiments (by Mikosch et al.):

“We note that the photoelectron spectrum extracted from the partial covariance image is noisier than the equivalent image obtained through background subtraction of the raw electron image. This is in part due to the nature of the covariance mapping procedure, which relies on statistical (Poisson) fluctuations in a noisy dataset and the detection of multiple particles, each of which have rather finite detection efficiencies. The influence of these factors on covariance mapping has been examined in detail recently. The covariant electron spectrum importantly contains information that cannot be gleaned from the raw photoelectron spectra, namely channel-resolved information by extracting photoelectron spectra correlated to the production of a given ion channel.”

5. I was wondering why He (and not Ne, or Ar) was used as a carrier gas in the experiments, although this leads to background signal close to the expected iodide signal.

We agree that Ne and Ar would have been good choices for carrier gas as well and we actually did take some data sets with argon and also without any carrier gas which both works for this molecule and the chosen experimental geometry. Helium was a default choice and turned out to be a valuable cross check and marker to identify the contributions, adjust the FEL irradiation for suppressed nonlinearity and also to establish and demonstrate the feasibility of the partial covariance method.

6. In figure 5 c), it looks as if the calculated 3Q0 and 1Q1 binding energy curves are shifted compared to Figure 8 and 9. Note: in Figure 8 the assignment for atomic iodine is wrong (excited, ground state interchanged). I was also wondering if any other excited states were considered, and if these could help to explain the current mismatch with the experimental spectrum at 300 fs (Fig. 5 b). For the interpretation of Figure 5 it might also have been nice to show a spectrum corresponding to UV off, i.e., corresponding to the ground state spectrum of the molecule (Figure 6). I assume that the I^{2+} signal was too low for the partial covariance analysis in the absence of the XUV pulse?

Indeed a slightly incorrect shift was applied to the simulated spectra in Figure 8 and 9 and we thank the reviewer for spotting this. We have corrected it in the revised manuscript. The assignment in figure caption 9 has been corrected as well. Currently, other excited states have not been considered. In the case of CH_3I it is rather well-established that the oscillator strength for other excited states is extremely low at around 267nm, but this is not well established for 1-iodo-2-methylbutane. We have added the following statement regarding this:

“Our theoretical work does not consider any possible contributions from other excited states, which, in the case of the related CH_3I molecule, are believed to have extremely low oscillator strengths at the pump energy used.”

As the reviewer correctly suggests, there is very little XUV-only I^{2+} signal. We have instead added to Figure 5a) a photoelectron spectrum correlated to production of I^+ , representing the signal from unpumped molecules. A remark has also been added to the caption of Figure 5 explaining this.

REVIEWERS' COMMENTS:

Reviewer #2 (Remarks to the Author):

I am satisfied with the authors review, and the manuscript can be now published in my opinion.

Reviewer #3 (Remarks to the Author):

I am extremely happy with the answers of the authors to my comments and their changes to the manuscript. This is a really interesting work and I enthusiastically recommend it for publication in Communications Chemistry.